# Reproducibility of the Wet Part of the Soil Water Retention Curve : A European Interlaboratory Comparison

Benjamin Guillaume[1], Hanane Aroui Boukbida[2], Gerben Bakker[3], Andrzej Bieganowski[4], Yves Brostaux[1], Wim Cornelis[5], Wolfgang Durner[6], Christian Hartmann[2], Bo V. Iversen[7], Mathieu Javaux[8], Joachim Ingwersen[13], Krzysztof Lamorski[4], Axel Lamparter[9], András Makó[10], Ana María Mingot Soriano[11], Ingmar Messing[11], Attila Nemes[12], Alexandre Pomes-Bordedebat[1], Martine van der Ploeg[3], Tobias Weber Karl David[13], Lutz Weihermüller[14], Joost Wellens[1], and Aurore Degré[1]

[1]Uliège - Gembloux Agro-Bio Tech, TERRA Teaching and Research Centre, Passage des Déportés 2, 5030 Gembloux, Belgium

[2]Instrumentation, Moyens Analytiques, observatoire en Géophysique et Océanographie (UAR IMAGO), Institut de Recherche pour le Développement (IRD)

[3]Wageningen University and Research, Netherlands

[4]Institute of Agrophysics, Polish Academy of Sciences, Poland

[5]Ghent University, Belgium

[6]Technische Universität Braunschweig, Germany

[7]Department of Agroecology, Aarhus University, Denmark

[8]UCLouvain, Earth and Life Institute, Belgium

[9]Federal Institute for Geosciences and Natural Resources, Germany

[10]Department of Soil Physics and Water Management, Institute for Soil Sciences, Centre for Agricultural Research, Herman Ottó street 15, 1022 Budapest, Hungary

[11]Swedish University of Agricultural Sciences, Sweden

[12]Norwegian Institute of Bioeconomy Research, Norway

[13]Institute of Soil Science and Land Evaluation, University of Hohenheim, Germany

[14]Agrosphere Institute IBG-3, Forschungszentrum Jülich GmbH, Germany

**Correspondence:** Benjamin Guillaume (benjamin.guillaume@uliege.be)

**Abstract.** The soil water retention curve (SWRC) is a key soil property required for predicting basic hydrological processes. SWRC is often obtained in laboratory with non-harmonized methods. Moreover, procedures associated to each method are not standardized. This can induce a lack of reproducibility between laboratories using different methods and procedures or using the same methods with different procedures. The goal of this study was to estimate the inter/intralaboratory variability of the measurement of the wet part (from 10 to 300 hPa) of the SWRC. An interlaboratory comparison was carried out between 14 laboratories, using artificially constructed, porous reference samples that were transferred between laboratories in according to a statistical design. The retention measurements were modelled by a series of linear mixed models using a Bayesian approach. This allowed the detection of sample-to-sample variability, interlaboratory variability, intralaboratory variability and the effects of samples changes between measurements. The greatest portion of the differences in the measurement of SWRCs was due to interlaboratory variability. The intralaboratory variability was highly variable depending on the laboratory. Some laboratories successfully reproduced the same SWRC on the same sample, while others did not. The mean intralaboratory variability over all laboratories was smaller than the mean interlaboratory variability. A possible explanation for these results is that all laboratories

used slightly different methods and procedures. We believe that this result may be of great importance regarding the quality of SWRC databases built by pooling SWRCs obtained in different laboratories. The quality of pedotransfer functions or maps that might be derived is probably hampered by this inter-/intralaboratory variability. The way forward is that measurement procedures of the SWRC need to be harmonized and standardized.

## 1    Introduction

Soil hydraulic properties control important hydrological processes such as infiltration, runoff and evapotranspiration (Assouline, 2021). The soil water retention curve (SWRC) is a soil specific hydraulic property that represents the relationship between the *matric potential* and the *water content* of the soil (Hopmans, 2019). The matric potential represents the energy state of water in soil, induced by physicochemical interactions between soil particles and water molecules (Luo et al., 2022). These physicochemical interactions are divided into capillary forces dominating at the wet part of the SWRC and adsorptive forces dominating at the dry part (Tuller et al., 1999; Tuller and Or, 2005). The wet part of the SWRC is considerably influenced by the soil pore network on a micrometer scale, which is affected by the so-called "soil structure". This highlights that the SWRC and hydrophysical behaviour of soils can be modified by management practices that influence its structure.

SWRCs are difficult, expensive and time consuming to obtain. SWRC data are therefore limited in space and time. The SWRC is obtained from the joint determination of a series soil matric potential and soil water content. Since the wet part of the SWRC is mostly determined by the distribution and connectivity of the largest pores ($> 1\mu$m), it must be measured *in situ* or in the laboratory on undisturbed soil samples. Soil water content can be measured by direct (gravimetric) method in the laboratory. To obtain matric potential, most laboratory methods impose a target matric potential on an undisturbed soil sample using an apparatus (Sand box (SB), Sand/Kaolinite box (SKB), Suction plate (SP), Pressure plate (PP)) (Klute, 1986; Dane and Hopmans, 2002; Mosquera et al., 2021). The sample is drained until its matric potential reaches equilibrium with the target matric potential. The SWRC can also be obtained via inverse modelling from an outflow experiment (One step outflow, Multi step outflow) (Hopmans et al., 2002). The SWRC can also be obtained by simultaneously measuring the water content and matric potential (with a tensiometer) of a soil sample evaporating in the free air and sealed at the bottom. Evaporation experiments also allow the soil hydraulic conductivity curve to be obtained simultaneously with the SWRC (Peters and Durner, 2008). This method can also be implemented *in situ* using tensiometers and water content sensors installed side by side (Zeitoun et al., 2021). The Kelvin equation may also be used to relate the relative humidity of the air in a closed chamber in vapor equilibrium with the soil water into a matric potential (Dew Point Hygrometer) (Gee et al., 1992).

Each method has its own accuracy and range of measurable matric potential. The determination of the SWRC over the full tension range (between saturation and wilting point or beyond) requires a combination of these methods. The comparison of these methods shows that they can lead to systematically different SWRCs for samples from the same soil (Bittelli and Flury, 2009; Schelle et al., 2013; Mosquera et al., 2021). The sources of variability are various and may relate to procedural factors, such as sample size (Ghanbarian et al., 2015; Silva et al., 2018). Apparent hydrostatic equilibria (broken hydraulic contact and water flow being stopped before reaching hydrostatic equilibrium) might occur with sand box, sand/kaolinite box, suction plate

or pressure plate methods, leading to overestimations of the water content, especially in the dry part of the SWRC (Madsen et al., 1986; Gee et al., 2002; Cresswell et al., 2008; Bittelli and Flury, 2009; Solone et al., 2012; Hunt et al., 2013; Schelle et al., 2013; de Jong van Lier et al., 2019). Contact materials are frequently used to improve the contact between the sample and the porous plate (Klute, 1986; Reynolds and Topp, 1993). The effects of these procedural aspects are not clearly established (Gee et al., 2002; Gubiani et al., 2013).

The methods that have been used, up to date, to measure the SWRC are different between laboratories, leading to non-harmonized datasets. Also, procedures for the same method differ from one laboratory to another. As a consequence, most SWRC databases that are used to create pedotransfer functions and maps pool non harmonized data from different laboratories (Wösten et al., 1999; Nemes et al., 2001; Weynants et al., 2013; Tóth et al., 2015, 2017). It is argued that an important source of uncertainty of pedotransfer functions comes from the uncertainty of measured input data and that the standardization of experimental protocols could significantly enhance their quality (Vereecken et al., 2010; Van Looy et al., 2017).

The Soil Program on Hydro Physics via International Engagement (SOPHIE), an independent initiative gathering European stakeholders in the field of soil hydrophysics, focuses on the harmonization and standardization of measurement of soil hydrophysical properties through international collaboration. To our understanding, no study other than that of Buchter et al. (2015) has carried out an interlaboratory comparison of SWRC measurements. This is partly due to the fact that an undisturbed soil sample cannot be transported from one laboratory to another and be measured several times without affecting the SWRC. Buchter et al. (2015) circumvented this problem by using many samples from the same location and only using the samples in one round of SWRC measurement. They demonstrated that soil heterogeneity in the sampling area was negligible compared to the variability introduced by the different sample extraction, preparation and analysis procedures. However, this approach becomes very difficult to achieve when soil samples have to be transported by air and to countries where importing soil is restricted. Thus, in addition to innovation in measurement techniques, SOPHIE is working on the development of artificially constructed reference samples and the organization of interlaboratory comparisons, starting with the SWRC.

This paper presents the results of first SOPHIE interlaboratory comparison for the measurement of the wet part (from 10 to 300 hPa) of the SWRC on reference samples. Fourteen laboratories participated in this study using their typical routine measurement methods and protocols. Four research questions were addressed:

1. What are the "intralaboratory" variabilities of the 14 participating laboratories?

2. What is the "interlaboratory" variability of the 14 participating laboratories?

3. Do reference samples made at different laboratories differ from each other in terms of water retention properties between 10 and 300 hPa?

4. Are the reference samples affected by time, measurements and/or transport between laboratories?

## 2 Materials and Methods

### 2.1 The reference sample

Each reference sample was composed by a mixture of 180 g of glass beads (0.250 mm < x < 0.500 mm), 20 g of pure air dry Portland cement and 35g of tap water. Once homogenized, the mixture was filled into a 100 cm$^3$ (5 cm height/ 5 cm diameter) stainless steel ring by frequently tapping it on a table to ensure that it was uniformly packed. The ring was closed at the bottom by a lid. Any excess material on the top was removed with a spatula. Each sample was allowed to cure for 72h at room temperature. The bottom lid was subsequently replaced with a Eijkelkamp nylon cloth supported by a rubber band. The sample, with the ring, the cloth and the rubber, was weighted. The empty ring, the cloth and the rubber were previously weighted separately.

### 2.2 The ring test

The ring test was organized in 14 soil physics laboratories. An example of a reference sample was sent to each laboratory alongside the material needed to construct five other samples. A total of 84 reference samples were constructed. The ring test consisted of three successive rounds of SWRC measurements. At the beginning of each round, each sample was initially saturated for 48h. The mass of each sample was then measured after equilibration at different matric potential (or suction) values : $\psi$ = 10 hPa ($\log_{10}(\psi)$ = 1.00), $\psi$ = 50 hPa ($\log_{10}(\psi)$ =1.70), $\psi$ = 100 hPa ($\log_{10}(\psi)$ = 2.00) and $\psi$ = 300 hPa ($\log_{10}(\psi)$ = 2.48). Equilibration times were 5 days at 10 hPa, 7 days at 50 hPa, 10 days at 100 hPa and 15 days at 300 hPa. Finally, samples were weighed after drying for 72 hours in an oven drying at 60°C. Gravimetric water content (wc in g.g$^{-1}$) was calculated by the ratio of water masses over dry masses (water content $= \frac{\text{fresh mass} - \text{dry mass}}{\text{dry mass}}$). The six samples from each laboratory were divided into three exchange modalities (Fig. 1); 2 samples were kept by the same laboratory all along the 3 rounds of measurements ("STAY"), two samples were sent to different laboratories between rounds ("MOVE") and the last two samples were sent to a different laboratory for the second round but were sent back to the original laboratory for the third round ("BACK"). This scheme was designed to estimate intralaboratory and interlaboratory variability as well as the effect of sample transfer between laboratories.

### 2.3 The data analysis

The final data set consisted of 250 SWRCs. Two curves were missing. Since each SWRC was composed of four successive measurement points whose relative value may depend on the previous point, the data were not independent. Statistical analyses were then based on parameter values of fitted functions. To model our dataset, a linear function with $\log_{10}(\psi)$ as the independent variable was adjusted to the measured wet part of SWRCs (Eq. 1).

$$wc_i = \beta_0 + (\beta_1 * (\log_{10}(\psi)_i - 1)) + \epsilon_i \tag{1}$$

Water content ($wc_i$) was linearly expressed as a function of $\log_{10}(\psi)$ values ($\log_{10}(\psi)_i$ = 1.00, 1.70, 2.00, 2.48). The $\log_{10}(\psi)_i -$ 1 was intended to set the first retention point at the intercept. The index $i$ represented the $i^{th}$ data point. $\beta_0$ and $\beta_1$ represented

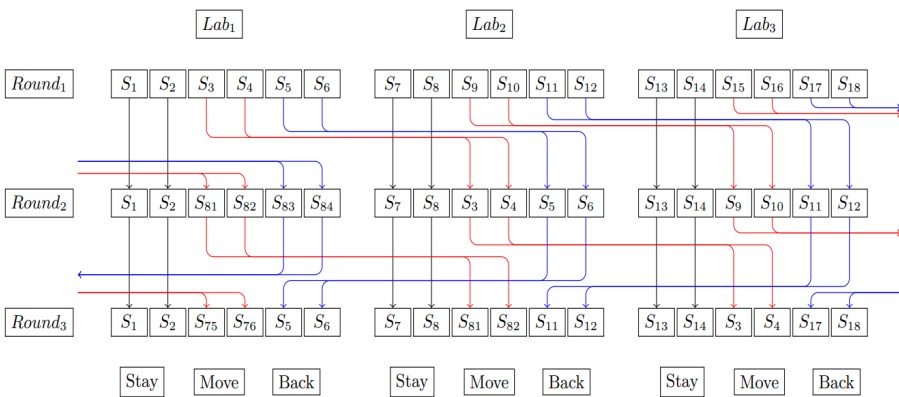

**Figure 1.** Example of sample exchange scheme of the ring test. Black arrows = STAY, red arrows = MOVE, blue arrows = BACK.

the mean intercept and the slope over all data. The term $\epsilon_i$ represented the residuals. The next step was to adjust a single linear model to each SWRC (Eq. 2).

$$wc_{in} = \beta_0 + \underbrace{z_{0n}}_{\text{varying intercept}} + \left( (\beta_1 + \underbrace{z_{1n}}_{\text{varying slope}}) * (\log_{10}(\psi)_i - 1) \right) + \epsilon_{in} \tag{2}$$

The $n$ index represented the $n^{th}$ SWRC. Depending on the modeled SWRC, intercepts ($z_{0n}$) and slopes ($z_{1n}$) were allowed to vary around a general intercept ($\beta_0$) and slope ($\beta_1$). This type of model refers to a linear mixed (effect) model.

The purpose of this study was also to investigate the interlaboratory variability as well as the differences between samples. Another linear mixed (effects) model was used to consider the by sample and by laboratory variability using adjustment terms called "random effects". The first random effect, $u_{0j}$ and $u_{1j}$ respectively adjusted $\beta_0$ and $\beta_1$ depending on the analysing

laboratory $j$ ($j \in [1, ..., 14]$). The other random effect, $v_{0k}$ and $v_{1k}$ respectively adjusted $\beta_0$ and $\beta_1$ depending on the sample $k$ ($k \in [1, ..., 84]$). This mixed effects model was described by the Eq. (3).

$$wc_{ijk} = \beta_0 + \underbrace{u_{0j} + v_{0k}}_{\text{varying intercept}} + \left( (\beta_1 + \underbrace{u_{1j} + v_{1k}}_{\text{varying slope}}) * (\log_{10}(\psi)_i - 1) \right) + \epsilon_{ijk} \tag{3}$$

Finally, the effect of sample changes between round 1 and round 3 on the intercept ($w_0$) and the slope ($w_1$) was modeled

through a "fixed effect" covariate. The covariate depended on a dummy variable associated to the round number; for the $1^{st}$ round, $round_i = -0.5$ and for the $3^{rd}$ round, $round_i = 0.5$. This later model (Eq. 4) was applied only to data associated with the "BACK" samples and "STAY" samples to avoid laboratory effects. The results were compared to determine whether the differences in measurements between rounds 1 and 3 were due to transport or differences caused by wear of the samples not

**Table 1.** Specificity of the participating laboratories in terms of device used (SB = Sand Box, SKB = Sand/Kaolinite Box, SP = Suction Plate, PP = Pressure Plate), contact material, cap on the samples during equilibriation periods, reference level used with respect to the sample at which the pressure was set, unit correction, plate cleaning procedure, saturation procedure and dry weight measurement procedures.

| Lab | Device | | | | Contact material | Cap | Pressure sample ref | Correction cm to hPa | Plate clean | Saturation | | Cooling after drying |
|---|---|---|---|---|---|---|---|---|---|---|---|---|
| | 10 hPa | 50 hPa | 100 hPa | 300 hPa | | | | | | Water type | Water level | |
| 1 | SB | SB | SB | SP | spheriglass 3000 | / | middle | / | water + brush | / | / | / |
| 2 | SB | SB | SB | SP | quartz meal | no | middle | no | no | demineralized | 45 mm | dessicator |
| 3 | SB | SB | SB | PP | no | no | middle | yes | H2O2 + water | demineralized | 25 mm | dessicator |
| 4 | SP | SP | PP | PP | milled sand | no | middle/unknown | no | water | demineralized | 47 mm | NA |
| 5 | SP | SP | SP | SP | filter paper | no | middle | yes | no | tap | 49 mm | dessicator |
| 6 | SB | SB | SB | PP | kaolinite on filter paper | no | middle | yes | light HClO solution | demineralized | 49 mm | no cooling |
| 7 | SB | SKB | SKB | SKB | no | yes | middle | yes | no | distilled | 50 mm | dessicator |
| 8 | SB | SB | PP | PP | no | yes | bottom | no | water + brush | deaerated | 50 mm | no cooling |
| 9 | SB | SB | SB | PP | loamy soil | no | middle | yes | water + brush | tap | 40 mm | dessicator |
| 10 | SP | SP | SP | SP | no | yes | bottom | no | water | distilled | 45 mm | NA |
| 11 | SP | SP | SP | PP | filter paper | yes | middle | no | water | demineralized | / | NA |
| 12 | PP | PP | PP | PP | no | no | middle | yes | water | demineralized | 40 mm | in the oven |
| 13 | SB | SB | SP | SP | sand | no | bottom | no | water | demineralized | 50 mm | in the oven |
| 14 | SB | SB | SB | PP | no | no | middle/bottom | no | tap water + brush | demineralized | 45 mm | NA |

related to transport.

$$wc_{ik} = \beta_0 + \underbrace{v_{0k} + w_0 * round_i}_{\text{varying intercept}} + \left( (\beta_1 + \underbrace{v_{1k} + w_1 * round_i}_{\text{varying slope}}) * (\log_{10}(\psi)_i - 1) \right) + \epsilon_{ik} \quad\quad (4)$$

All parameters from each models were estimated using Bayesian statistics. Posterior distributions were sampled with a Markov Chain Monte Carlo (MCMC) algorithm implemented in C++ through an R package called "RStan" (Carpenter et al., 2017). Priors are noninformative; i.e. centered normal distributions for random effects parameters and uniform distributions for general intercept, general slope, fixed effect covariates and variance parameters. Sensitivity analyses of priors and validations of models

were also carried out. Inference was based on Bayes factors and Bayesian Credible Intervals of posterior distributions. More details are available in the Supplementary Materials (doi : ).

## 3 Results

### 3.1 Procedures of laboratories

Each laboratory received the same procedure to measure the SWRC. However, it allowed some freedom and some laboratories

did not perfectly implement it. For instance, laboratory 8 dried the samples at 100°C instead of 60°C. Hence, laboratories used slightly different procedures as shown in Table 1. Laboratories mostly used the Sand Box (SB) at 10, 50 and 100 hPa. At 300 hPa, the Suction Plate (SP) and the Pressure plate (PP) were dominating. Lab 7 was the only one to use the Sand/Kaolinite Box (SKB).

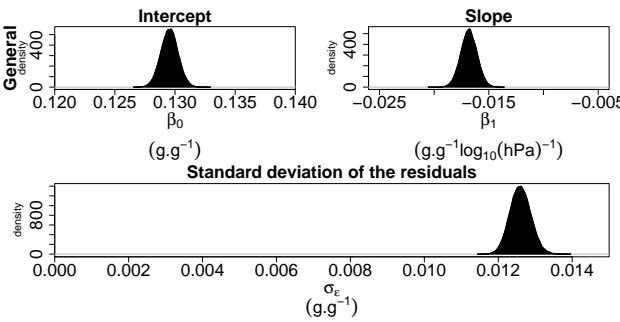

**Figure 2.** Densities of the posterior probability distribution of the general intercept, $\beta_0$, the general slope, $\beta_1$, and the standard deviation of the residuals, $\sigma_\epsilon$.

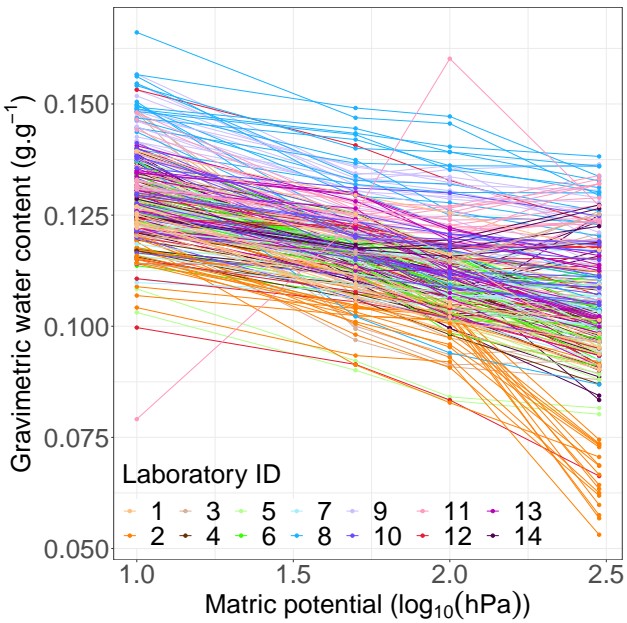

**Figure 3.** Observed series of four successive retention points (10 hPa, 50 hPa, 100hpa and 300 hPa) connected by straight lines. Water content is gravimetric (g.g$^{-1}$). One colour represents one laboratory. This colour code is kept constant throughout the paper.

### 3.2 The simple linear model : SWRCs are very variable

The simple linear regression (Eq. 1) with the $\log_{10}(\psi)$ as predictor was used to model the data set. The posterior probability distribution of the general intercept ($\beta_0$), slope ($\beta_1$) and the standard deviation of the residuals ($\sigma_\epsilon$) are shown in Fig. 2. The mean value of $\sigma_\epsilon$ was fairly high ( 0.0126 g.g$^{-1}$). Indeed, as shown in Fig. 3, the variability of measured SWRCs was large (spreading of the curves). The following steps were devoted to explaining the origin of this variability.

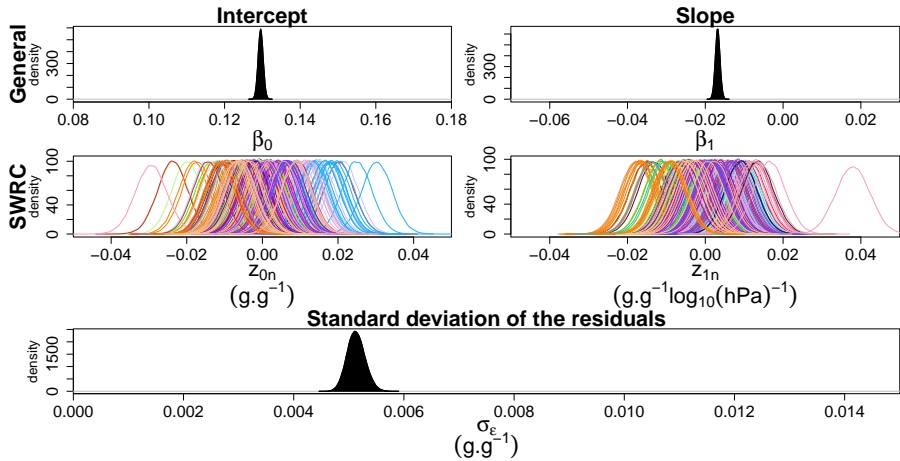

**Figure 4.** Densities of the posterior probability distribution of the general intercept, $\beta_0$, the varying intercept, $z_{0n}$, the general slope, $\beta_1$, the varying slope, $z_{1n}$, and the standard deviation of the residuals, $\sigma_\epsilon$. Densities of the posterior probability distribution of individual SWRC parameters ($z_{0n}$ and $z_{1n}$) are colored as function of the analyzing laboratory.

### 3.3 A linear model for each SWRC to estimate the intralaboratory variability

The next step was to model a single linear regression for each SWRC (Eq. 2). The posterior probability distribution of the general intercept ($\beta_0$), slope ($\beta_1$), the individual intercept ($z_{0n}$) and slope ($z_{1n}$) and the standard deviation of the residuals ($\sigma_\epsilon$) are shown in Fig. 4. The intercept ($z_{0n}$) and slope ($z_{1n}$) parameters are different for each individual SWRC. These individual parameters explain the variability that exists between all SWRC. Hence, the mean value of $\sigma_\epsilon$ presented in Fig. 4 decreased by approximately 60 % compared to the previous model (Fig. 2) and now only represents a fitting error introduced by the choice
of modeling SWRCs by linear regressions.

From these results, one can determine the standard deviation of $z_{0n}$ and $z_{1n}$ for each of the "STAY" samples (between the 3 rounds). As each laboratory measured two "STAY" samples, an estimate of the intralaboratory variability of each laboratory can be made by pooling the density estimates of the standard deviation of the two samples (Fig. 5). As the intralaboratory variability associated with each of the two "STAY" sample is determined separately and are then merged together, this allows to
minimize the effect of possible variations between the two "STAY" samples. Intralaboratory variability is therefore defined as the variability between retention curves, modeled by linear regressions, measured on a similar ("STAY") sample within a same laboratory that uses a given measurement procedure.

The estimate of the mean intralaboratory standard deviation of all laboratories pooled together (Fig. 5 bottom row) was 0.00533 g g$^{-1}$ (95% Credible intervals (CrI) 0.00018-0.01138 g g$^{-1}$) for the intercept and 0.00519 g g$^{-1}$log$_{10}(hPa)^{-1}$ (95%
CrI 0.00038-0.01068 g g$^{-1}$log$_{10}(hPa)^{-1}$) for the slope (Table A1). Figure 5 (top row) also shows that the intralaboratory variability was quite different depending on the laboratory. Some laboratories succeeded in repeating similar SWRCs results on a same sample while others failed.

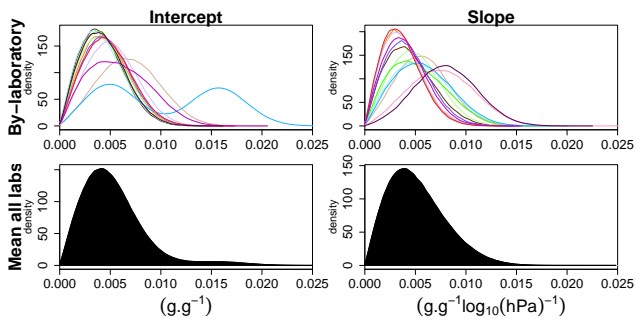

**Figure 5.** Densities of the posterior probability distribution of the varying intercept, z0n, and the varying slope, z1n, standard deviation of the two "STAY" samples of each laboratory (top row) and all laboratories together (bottom row).

## 3.4 What is the interlaboratory and sample variability?

Although all laboratories were given the same procedure to build the reference samples, the conditions under which they were constructed differed between laboratories. Hence, the bulk density of samples at the beginning of the experiment was variable depending on the laboratory that constructed the sample (Table A2). Indeed, the difference between the mean bulk density of samples constructed by the lab 1 (highest bulk density) and lab 14 (lowest bulk density) was 0.1573 g.cm$^{-3}$. Hence, the later linear mixed (effect) model was used to estimate the interlaboratory variability on the SWRC considering the differences between samples (Eq. 3). Densities of the posterior probability distribution of the general intercept ($\beta_0$) and slope ($\beta_1$), the random effect of laboratory on the intercept ($u_{0j}$) and slope ($u_{1j}$), the random effect of sample on the intercept ($v_{0k}$) and slope ($v_{1k}$) and the standard deviation of the residuals ($\sigma_\epsilon$) are shown in Fig. 6. The mean value of $\sigma_\epsilon$ presented in Fig. 6 decreased by approximately 40 % compared to the simple linear model (Fig. 1). Indeed, a part of the variability between SWRCs has been explained by sample and laboratory random effects. Parameter values of the laboratory random effects ($u_{0j}$ and $u_{1j}$) show how SWRCs systematically deviate depending on the analysing laboratory. Differences between samples were also estimated with parameter values of the samples random effects ($v_{0j}$ and $v_{1j}$). The wider dispersion of the laboratory random effect parameters indicates that the analysing laboratory explained a larger proportion of the overall variance than the analysed sample. Indeed, on the intercept, the mean laboratory random effect standard deviation ($\sigma_{u0}$) was 0.00872 g g$^{-1}$ while it was 0.00350 g g$^{-1}$ for the sample random effect ($\sigma_{v0}$). The same observation applies to the slope with a mean standard deviation of 0.00602 g g$^{-1} \log_{10}(hPa)^{-1}$ for the laboratory random effect ($\sigma_{u1}$) and 0.00451 g g$^{-1} \log_{10}(hPa)^{-1}$ for the sample random effect ($\sigma_{v1}$). The mean laboratory random effect standard deviations on the intercept and slope values ($\sigma_{u0}$ and $\sigma_{u1}$) represent an estimation of the interlaboratory variability.

It should be noted that results presented here are only representative of the reference samples that were measured in this particular case. Thus, estimates of interlaboratory (and intralaboratory) variability values are not directly transferable to other samples of different nature with different retention characteristics.

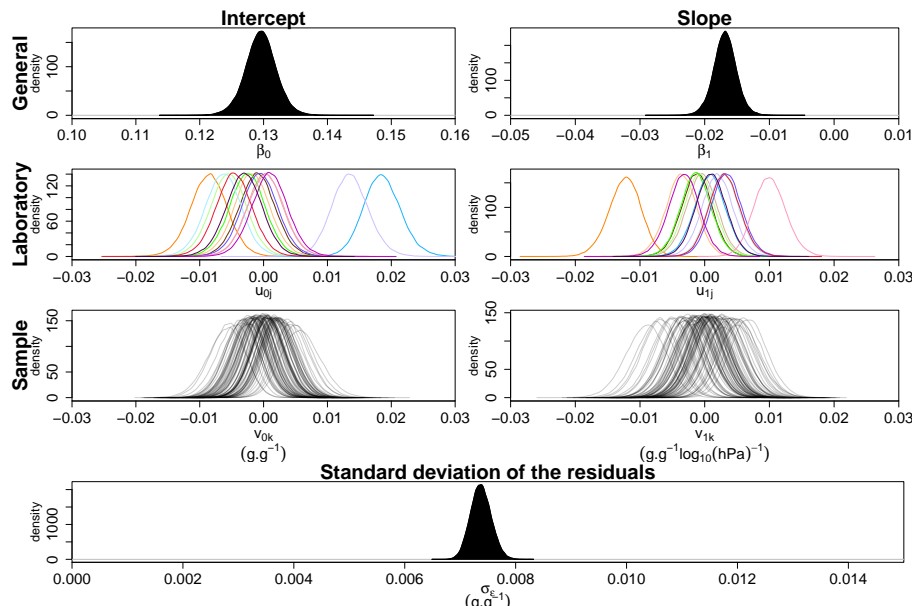

**Figure 6.** Densities of the posterior probability distribution of the general intercept ($\beta_0$), the random effect of laboratory on the intercept ($u_{0j}$), the random effect of sample on the intercept ($v_{0k}$), the general slope ($\beta_1$), the random effect of laboratory on the slope ($u_{1j}$), the random effect of sample on the slope ($v_{1k}$) and the standard deviation of the residuals ($\sigma_\epsilon$).

## 3.5 Do the samples change between rounds?

In order to assess the effect of possible sample changes on the SWRCs measurements, a last model was separately fitted to the data from "BACK" and "STAY" samples (Eq. 4). The Bayes factor indicated that the predicted data of "BACK" samples were 46.60 times more probable under the model that takes the round effect into account than the model without the round effect. Moreover, the 95% credible interval of the posterior probability distribution of the "round" effect (Fig. 7) laid outside 0 for the intercept ($w_{0b}$) and the slope ($w_{1b}$). Therefore, "BACK" samples changed between round 1 and round 3. On the other hand, the Bayes factor indicated that the predicted data of "STAY" samples were 4348 times less probable under the model that takes the round effect into account than the model without the round effect. This suggests that the changes in the retention properties of the "STAY" samples between the first and third rounds were negligible. In contrast, the 95% credible interval of the posterior probability distribution of the "round" effect (Fig. 8) suggests that the "STAY" samples changed between round 1 and round 3 as it laid outside 0 for the intercept ($w_{0s}$) and slope ($w_{1s}$). Thus, for the "STAY" samples, the Bayes factor and 95% credible interval yielded two opposite conclusions. Also, the dry mass of "STAY" samples increased between round 2 and 3 (p = 0.016*) which was not the case with "BACK" samples (p = 0.199 ns). The round effect on "STAY" samples is discussed in section 4.3.

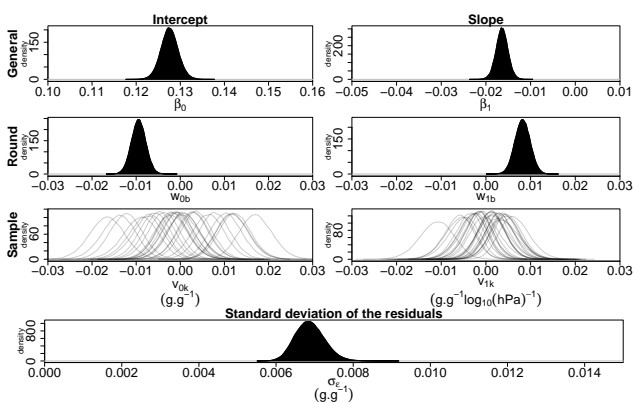

**Figure 7.** Densities of the posterior probability distribution of the general intercept ($\beta_0$), the fixed effect of the transport on the intercept ($w_{0b}$), the random effect of sample on the intercept ($v_{0k}$), the general slope ($\beta_1$), the fixed effect of the transport on the slope ($w_{1b}$), the random effect of sample on the slope ($v_{1k}$) and the standard deviation of the residuals ($\sigma_\epsilon$). The model is applied to the "BACK" data for the $1^{st}$ and the $3^{rd}$ rounds.

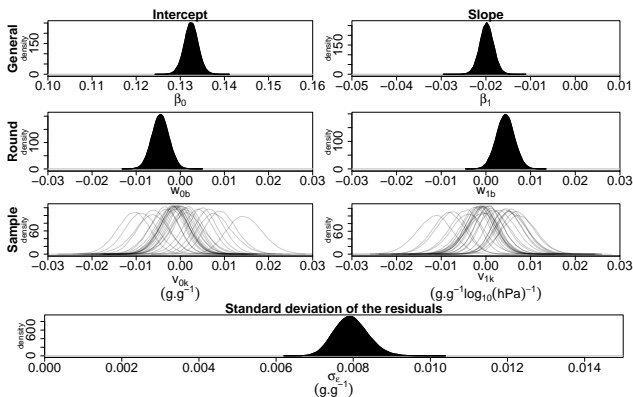

**Figure 8.** Densities of the posterior probability distribution of the general intercept ($\beta_0$), the fixed effect of the transport on the intercept ($w_{0s}$), the random effect of sample on the intercept ($v_{0k}$), the general slope ($\beta_1$), the fixed effect of the transport on the slope ($w_{1s}$), the random effect of sample on the slope ($v_{1k}$) and the standard deviation of the residuals ($\sigma_\epsilon$). The model is applied to the "STAY" data for the $1^{st}$ and the $3^{rd}$ rounds.

## 3.6 Increasing SWCRs

57 of the 250 measured SWRCs showed an increase in water content between at least two increasing suction steps (Fig. 3). Whatever the origin, this increase of water content is physically impossible. It appeared that the occurrence of these anomalies depended on the analysing laboratory, with some having no anomalies and others having a large number of occurrences. Indeed, laboratories 3, 11 and 14 together accounted for 35 of the 57 anomalies recorded (Table A3). Moreover, this anomaly has happened more than once for some samples such as for the samples 1, 2, 11, 15, 56, 61, 62, 63, 65 and 66. The samples 15,

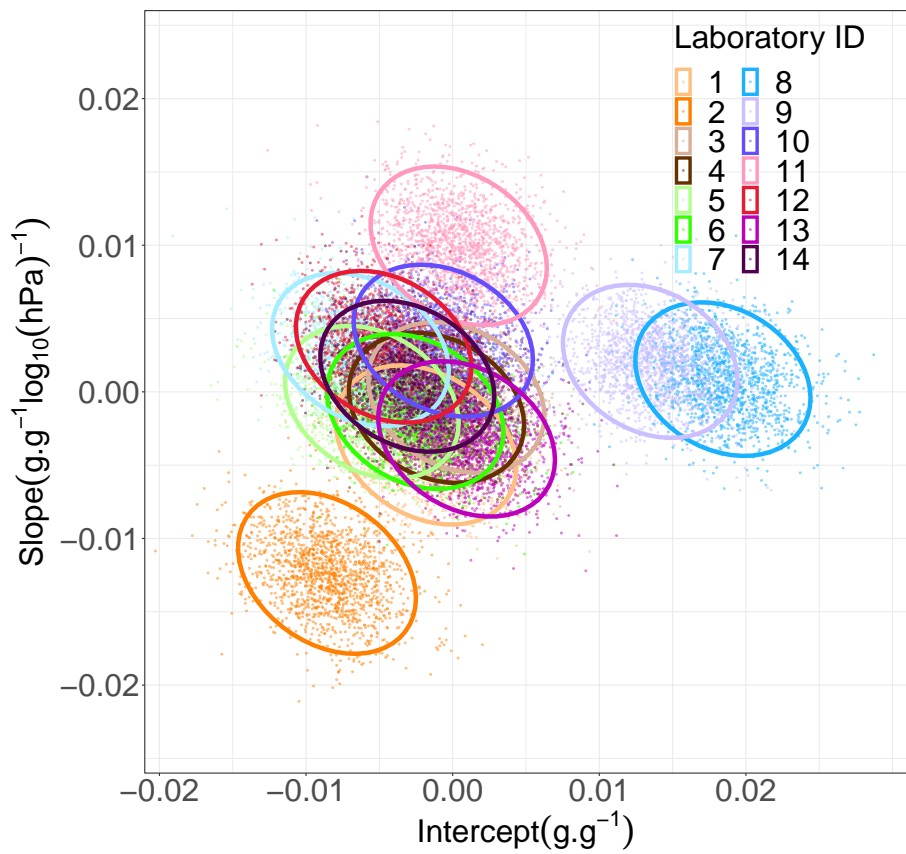

**Figure 9.** Joint posterior probability distribution of the laboratory random effect on the intercept ($u_{0j}$) and slope ($u_{1j}$).The ellipses are for illustrative purposes only. Please, refer to Table A1 for estimates of laboratory random effects.

63, 65 and 66 showed this anomaly in two different laboratories. Also, it occurred more often the drier the sample was. There
were 55 occurrences between 100 and 300 hPa while there were only 9 between 50 and 100 hPa and 3 between 10 and 50 hPa.
For some SWRCs, there was more than one occurrence.

## 4 Discussion

### 4.1 interlaboratory variability

This study confirms that there are systematic differences in the measurement of SWRCs depending on the laboratory (Fig. 6).
This is true even for laboratories using similar devices (eg. lab 6 vs 9). These systematic differences in the measurement of
SWRCs attributed to laboratories resulted in a large interlaboratory variability. The portion of variability attributed to differ-

ences between laboratories was larger than the portion of variability attributed to intrinsic differences between samples. This is concerning since it was shown, through the comparison of the bulk densities (Table A2), that the samples were different even at the very beginning of the experiment. From saturation to drying, all laboratories used slightly different procedures (Table 1).

Similarly to the argument of Buchter et al. (2015) on the measurement of macropore volume, total pore volume and saturated hydraulic conductivity, we believe that procedural differences between laboratories can be at the origin of this interlaboratory variability. The identification of the aspects of the procedures that influence SWRCs measurements is challenging since these were not studied in isolation. This is a multidimensional problem that remains beyond the scope of this article. Nonetheless, an attempt is made to hypothesize the effect of some of these procedural aspects. It should also be mentioned that the true value

of the SWRC was unknown. Laboratories were compared according to their relative position with respect to the others and not against a fixed target value.

Differences between laboratories are unlikely to be associated with differences in the analysed samples. The intrinsic differences between samples was considered by the model (Eq. 3) using the sample random effect. Indeed, there was no correlation between the intercept parameter of each laboratory and the average bulk density of the samples analysed by each laboratory (r

= -0.0078).

A first possible source of this interlaboratory variability could be attributed to the different devices used. To our knowledge, no study has yet attempted to compare SWRCs obtained with SB, SKB, SP or PP. Nonetheless, Schelle et al. (2013) found that SWRCs measured with SP were less reproducible (wider spread) than those measured with the evaporation method in the $\psi$ range of 0-300 hPa. This wider spread could be associated with contact issues with the plate and/or with the smaller sample size

used with the SP method which could be smaller than the representative elementary volume of the soil. They also found that, for sandy soils, water contents are systematically smaller for SWRCs obtained with SP than with the evaporation method. These differences could be due to smaller initial saturation degrees of the sandy samples measured with the SP compared to those measured with the evaporation method. Temperature effects and dynamic non-equilibrium effects as found by Diamantopoulos and Durner (2012) could also have played a role in these differences. Buchter et al. (2015) also compared macropore volume

(as equal to the difference between the total pore volume or water content at saturation and water content at a matric potential of 60 hPa) obtained with SB and with pressure cells on real soil samples. It can be observed that the volume of macropores determined by laboratories using the SB is generally greater with a greater variability than when determined by laboratories using the pressure cell. However, it is not clear from their results whether this difference is due to the method of obtaining the water content at 60 hPa (SB or pressure cell) or to the method of estimating the total pore volume (from bulk density, weighting

at saturation, etc) which was also different between laboratories.

Moreover, all laboratories used two different devices between 10 and 300 hPa, except labs 5, 10 and 12 that kept the same device for each pressure step. Switching from a suction to pressure system may affect the measurement of the SWRC. In a suction device, the suction is applied by a hanging water column via a continuum of water. In a pressure device, the pressure is applied via the air to the soil and plate water, while the plate bottom is at atmospheric air pressure. So, the propagation of the

applied tension/pressure might be different.

The procedures for the dry mass measurement of the samples may also have played a role in the observed differences. Indeed, the estimation of the intercept parameter of laboratory 8, which dried the samples at 100°C, was higher than the ones from the other laboratories, which dried the samples at 60°C (Fig. 6). This suggests that the dry masses measured by laboratory 8 were lower than those measured by the other laboratories. This may be explained by the fact that the assumed "zero" water content corresponds to two different water content values associated with two different water potentials. Indeed, the equilibrium potential of the water contained in a sample after drying depends on the drying temperature and the relative humidity of the air inside the oven (Ross et al., 1991). At liquid/vapor equilibrium and at constant relative humidity, the same sample dried at 60°C will have a higher water content than if it was dried at 100°C. This difference in water content will also depend directly on the shape of the dry part of the retention curve. Furthermore, for the same sample at the same initial condition, the drying time required to remove the same amount of water increases as the drying temperature decreases. If liquid/vapour equilibrium has not been reached after the prescribed time of drying, it is possible that the amount of water released is lower when drying at 60°C than at 100°C.

Another possibility to explain differences is the way laboratories maintained hydraulic contact between the draining porous media and the sample, enabling water to be released from the sample until hydrostatic equilibrium is reached. When the draining porous media was rigid (eg. ceramic) some laboratories used a "contact material" to improve the hydraulic contact (Table 1). From this study, it cannot be concluded that the use of contact materials did or did not improve the hydraulic contact between the samples and the porous plates. The results of laboratory 11 suggest that contact issues may have occurred even if filter paper was used as the contact material.

Nevertheless, the use of contact materials may sometimes be useful when considering laboratories using the same devices. Hence, it appears that the use of filter paper by laboratory 5 resulted in more water being released (more negative slope) than laboratory 10, which did not use any contact material but used the same devices (Fig. 9). Gubiani et al. (2013) also found that filter paper allowed more water to be released than polyester fabric and synthetic knitwear at 5000 and 15000 hPa with the PP. The use of kaolinite by lab 6 and loamy soil by lab 9 as contact material seems to yield in more water being released between 100 and 300 hPa than laboratories 3 and 14 that did not used any contact material but have used the same devices (results not shown). However, when looking at the entire domain of suctions (from 10 to 300 hPa) the effect of kaolinite or loamy soil was negligible (Fig.9). Gee et al. (2002) also found kaolinite ineffective in speeding equilibrium (or increasing hydraulic conductance), with inconsistent effects, at 15 000 hPa. Further work should be done to determine which contact materials are useful depending on the specific situation.

An option to check if the hydrostatic equilibrium is achieved is to connect the porous drainage medium to a graduated cylinder and monitor the water the amount of water drained from the sample. Once no more water flowing out of the sample is observed, hydrostatic equilibrium is considered to be achieved. This setup has to be sealed in order to ensure that there is no evaporation. The advantage of such a system is that one does not need to assume the equilibration time a priori. To our knowledge, this setup was used by laboratory number 8. However, it is still possible, with this setup, that the hydraulic contact is broken and the flow of water is stopped before hydrostatic equilibrium is reached which refers to as an apparent hydrostatic equilibrium.

It should also be mentioned that with devices using a hanging water column as suction regulation system, the applied suction is usually expressed in cm of water column. Units hPa and cm are commonly considered equivalent, but in fact 1 cm of vertical water column corresponds to 0.98 hPa. This is usually overlooked when units are transformed (cf. Table 1). This bias may constitute a small part of the variability between laboratories and calls for harmonization of units.

In addition, the reference level compared to the sample at which the suction is applied varies between laboratories and devices used. Some laboratories applied the prescribed suctions to the bottom of the samples while others applied it to the middle (cf. Table 1). Laboratories that applied suction to the bottom of the samples systematically applied 2.5 cm more suction than those that applied it to the middle. This difference could have easily been corrected for after the measurement. However, in practice, this information is never transmitted from the laboratory to the end user. Hence, when pooling data from different laboratories using different reference levels to apply suction, this unknown difference introduces variability.

There might be other procedural aspects that can be responsible for these differences between laboratories (saturation procedure, porous plate maintenance during the experiment, means of preventing air leakages and evaporation, maintenance of the ceramics and the sandboxes, weighting procedure, maintenance of the scales, etc.)(Table 1). Big errors can be avoided by a quality check of the results. To our knowledge, only one laboratory used a reference sample to control the quality of their SWRCs measurements as a standard operating procedure.

## 4.2   intralaboratory variability

Some laboratories successfully reproduced SWRCs of a same ("STAY") sample while others failed (Fig. 10). Nevertheless, the estimate of the mean intralaboratory variability over all laboratories was smaller than the mean interlaboratory variability, but was more uncertain since it was drawn from less samples and since the intralaboratory variability was quite different between laboratories. Obviously, this variability can partly be attributed to the different methods and procedures that existed between laboratories that were discussed above. Some procedures ensured fairly good repeatability of results while others did not.

The two laboratories with the greatest intralaboratory variability on the slope (cf. Fig. 5 & 10: light pink and dark purple curves) were also among those with the most anomalies (cf. Table A3: lab 11 and 14). Concerning laboratory 8 (cf. Fig. 5: cyan curve), the bimodal shape of the intralaboratory variability on the intercept shows that for one sample the variability was high while it was low for the second. This bimodality clearly indicates that the estimation of individual intralaboratory variabilities are rather uncertain as they are only based on two samples measured with only three repetitions. This calls for further trials on reference samples to obtain a more reliable estimate of intralaboratory variability. Nevertheless, this provides an insight into the way forward to improve data quality management in soil physics laboratories.

## 4.3   Effect of repeated measurements and/or transport on the samples

It appears that there was a slight effect of the transfer between laboratories on the "BACK" samples. The values of $w_{0b}$ and $w_{1b}$ indicate that SWRCs of "BACK" samples globally have a smaller intercept with a flatter slope after being transported (Fig. 7). This pattern might indicate a shift to more small pores. A possible explanation for these changes in porosity is the calcium carbonation of the cement. This reaction ($Ca(OH)_2 + CO_2^{atm} \rightleftharpoons CaCO_3 + H_2O$) forms $CaCO_3$ precipitates inside

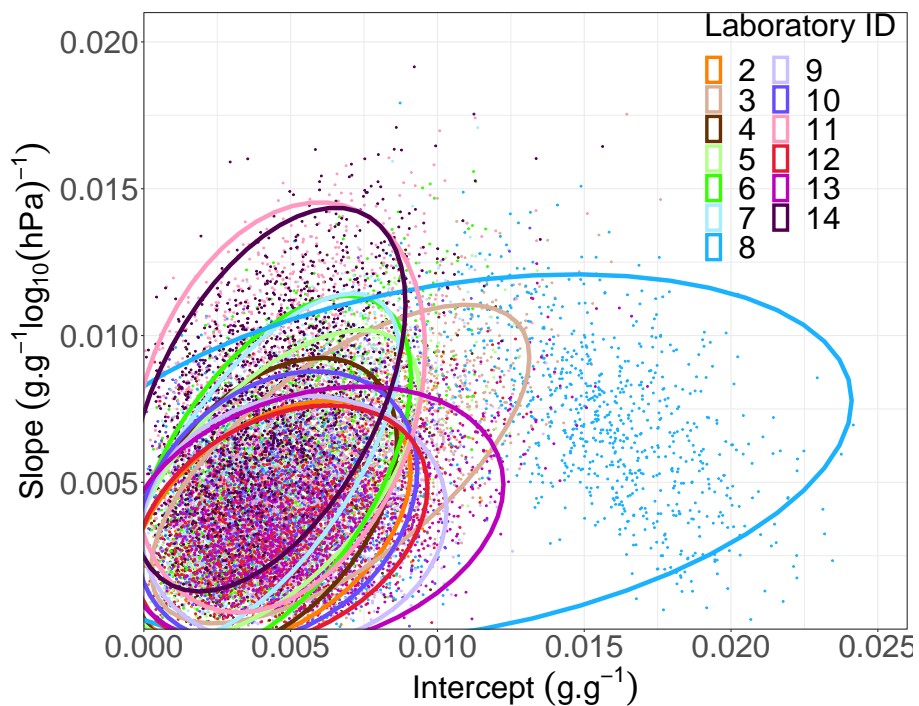

**Figure 10.** Joint posterior probability distribution of the varying intercept, $z_{0n}$, the varying slope, $z_{1n}$, standard deviation of the two "STAY" samples of each laboratory (intralaboratory standard deviations). The ellipses are for illustrative purposes only. Please, refer to Table A1 for estimates of intralaboratory variabilities

the pore network inducing a shift of the pore size distribution towards smaller pores, a decrease of the total porosity, pore clogging and a loss of pore connectivity in cement based materials (Šavija and Luković, 2016; Auroy et al., 2015). This hypothesis is also motivated by the fact that the dry masses of the samples increased significantly between rounds 2 and 3 for the samples "STAY" and the dry masses did not decrease significantly for the samples "BACK" even if losses of materials were reported by the laboratories. Indeed, Houst (1993) estimated that the carbonation induced increase in bulk density (due to $CO_2$ fixation) from a non carbonated to a fully carbonated cement paste was 1.60 to 2.03 g.cm$^3$. However, the actual contribution of this phenomenon to changes in the retention properties of each reference sample is difficult to estimate, as the degree of carbonation was influenced by environmental factors ($CO_2$ concentration, air humidity, water content of the cement, etc.) which have not been controlled. Nevertheless, this significant transport effect could have led to an overestimation of the

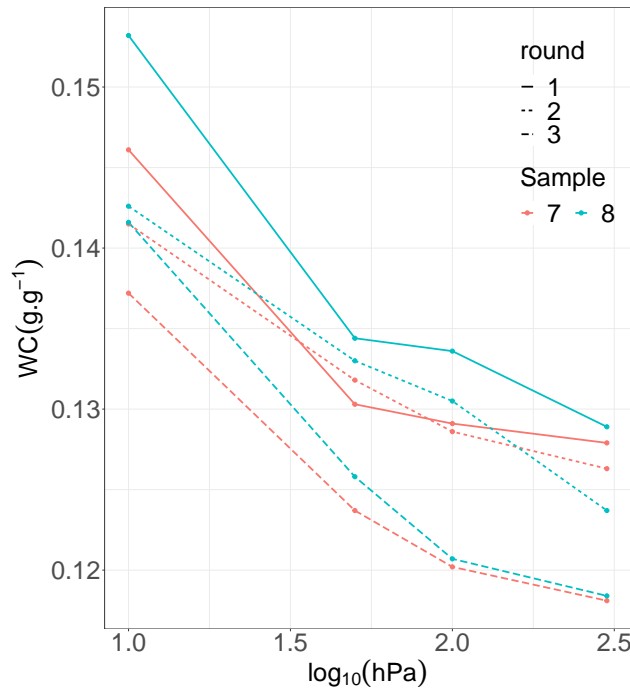

**Figure 11.** SWRCs of the "STAY" samples measured by lab 9.

interlaboratory variability, as a part of the variability of the SWRC measurements can be attributed to sample changes. The use of cement to construct such reference samples should certainly be avoided in the future.

Although it was not significant in general for "STAY" samples, some laboratories still seem to report sample changes be-
325 tween rounds. The changes followed the same patterns as for "BACK" samples, which were significant it that case (Fig. 8). Indeed, for "STAY" samples of laboratories 2, 3, 4, 6, 9, 10, 11, 12 and 13 it seems that the water content at 10 hPa ($\log_{10}(\psi)$ = 1) systematically decreased with time. This was particularly visible for lab 9 (Fig. 11). The origin of the changes of the "STAY" samples can partly be attributed to the same origin as for the "BACK" samples. The degree of changes may therefore be influenced by the way the samples were handled and stored, resulting in less wear for non transported samples. Neverthe-
330 less, the wear of the "STAY" samples implies that the estimation of the intralaboratory variability was certainly inflated as it included the variability attributed to sample changes between rounds. It should be mentioned that for some laboratories with the highest "intralaboratory" variability (laboratories 8 and 14), this trend was not visible, indicating that for these laboratories the variability attributed to procedures was probably more important than the variability attributed by sample changes.

## 4.4 Outliers

Many reasons might be elicited to explain the fact that some SWRCs showed an increase in water content between at least two increasing suction steps (Fig. 3). Obviously, this happened depending on a combination of reasons related to the laboratory but also to the sample being analyzed (Table A3).

A possible reason can be the lack of hydraulic contact between the draining porous media and the sample, preventing water to be released in time from the sample. This is supported by the higher frequency of outliers when the sample was drier, as hydraulic conductivity decreases as the sample dries (Gee et al., 2002). Indeed, there is a possible scenario in which samples may absorb water but may not be able to release it according to the driving (higher) pressure. Measurements in a pressure chamber typically involve placing samples on pre wetted ceramic plates. However, especially when a wet contact material is used, a unsaturated sample may start absorbing water (from the plate and the contact material) and resaturate before the chamber is pressurized. Once the chamber is pressurized the excess of water may not be drained if the hydraulic contact is not well established. Hydraulic contact could have been hampered by the rigid nature and non flat bottom topography of reference samples which did not fit the porous plate or by the use of shrinkable contact materials. When using the pressure plate, it is also possible that a "backflow" of water from the ceramic to the sample may occur between the release of the pressurized air and the disconnection of the sample from the plate (Richards and Ogata, 1961). Nevertheless, increasing SWRC also occurred with sandboxes and sand/kaolinite boxes, where the applied suction was not released when the sample was disconnected.

## 4.5 A way forward to further improve the quality of SWRCs measurements

The results presented in sections 3.3 and 3.4 show that interlaboratory and intralaboratory variability exists in the measurement of the SWRC. As discussed in Section 4.1, we suspect that some part of this variability can be attributed to the different methods and procedures used by each laboratory. Therefore, this variability could potentially be reduced by improving procedures and methods. Ideally, these should be adapted in such a manner that they allow the closest possible estimation of the "actual" SWRC of any soil sample. The prerequisite for this to be feasible is to have a fixed point of comparison; i.e. a reference sample whose retention properties are well known and remain relatively stable over time. However, these two requirements were not fulfilled in this study. Therefore, the way in which procedures and methods should be adapted remains an open question. Nevertheless, based on the results of this study, we can suggest some general future directions to further improve the quality of the laboratory SWRCs measurements.

We believe that the reproducibility of SWRCs measurements within a same laboratory would be improved if a reference sample was used by each laboratory as an internal quality control. This implies that the SWRC of the reference sample must remain relatively stable over time, but its true value should not necessarily be known a priori. The reference samples used in this study are already used by one laboratory as an internal quality control. According to their experience, the samples remain fairly stable for about ten measurements as long as they are not oven dried.

Also, an important element that would allow to move forward would be to have a point of comparison. Thus, the emphasis should be placed on the development of reference samples for interlaboratory comparisons. Internal trials are underway to

build reference samples with clays or sintered glass beads. Also, reference samples based on parallel capillary bundles with adjustable diameters could allow a "theoretically" reference retention curve to be calculated from Jurin's law. The feasibility of developing such a reference sample is also being explored.

The interlaboratory and intralaboratory variability could be reduced by improving and standardizing procedures and harmonizing methods and data. Ideally, all laboratories should endorse a unique Standard Operational Procedure for the same method and methods should be harmonized between each other. Improving and standardizing procedures requires a full assessment of the effect of each step of the procedures on the final SWRC measurement on a reference sample that fulfills the above-mentioned conditions (a priori known and relatively stable). Harmonization of methods can be achieved with interlaboratory

comparisons on the same types of reference samples.

    Finally, since procedures and methods could have an impact on the final measurement of the SWRC, the transparency of the procedures and methods used in SWRCs datasets should be ensured.

## 5   Conclusions

Here, we presented an interlaboratory comparison of the measurement of the wet part of the SWRC conducted between 14

laboratories using artificially constructed, structured and porous samples as references. The experimental design combined with the data analysis procedure allowed the inter- and intra-laboratory variability to be revealed. Systematic differences in the measurement of SWRCs attributed to laboratories resulted in a large interlaboratory variability. The variability explained by the differences between laboratories was more important than the variability explained by intrinsic differences between samples. The intralaboratory variability was laboratory dependent. The mean intralaboratory variability over all laboratories

was approximately 45% smaller on the intercept and 15% smaller on the slope than the mean interlaboratory variability (Table A1). The samples slightly changed during the interlaboratory comparison, inducing variability which was part of our estimate of intra/interlaboratory variabilities. We believe that another part of the intra/interlaboratory variability can be attributed to the different methods and procedures followed by each laboratory that were not standardized. This calls for standardization of procedures and harmonization of methods. This should be performed in the light of a fixed target value: a reference sample

whose retention properties are well known and preferably remain stable over time. We believe that without such an effort, pedotransfer functions and large scale maps of soil properties produced with databases constructed on multiple laboratories' inputs will keep carrying unknown levels of uncertainty and bias.

*Code and data availability.*   Data and R code will be available on an online and open access repository linked from the manuscript through a DOI.

**Table A1.** Summary table of interlaboratory and intralaboratory variability results.

| Lab | Laboratory random effect on the intercept, $u_0$ (g.g$^{-1}$) | | | Laboratory random effect on the slope, $u_1$ (g.g$^{-1}$.$\log_{10}(hPa)^{-1}$) | | | Intralab SD on the intercept (g.g$^{-1}$) | | | Intralab SD on the slope (g.g$^{-1}$.$\log_{10}(hPa)^{-1}$) | | |
|---|---|---|---|---|---|---|---|---|---|---|---|---|
| | Mean | 95% CRI | | Mean | 95% CRI | | Mean | 95% CRI | | Mean | 95% CRI | |
| 1 | **-0,00184** | -0,00780 | 0,00396 | **-0,00363** | -0,00851 | 0,00110 | **/** | / | / | **/** | / | / |
| 2 | **-0,00855** | -0,01450 | -0,00278 | **-0,01237** | -0,01770 | -0,00762 | **0,00460** | 0,00047 | 0,00877 | **0,00377** | 0,00032 | 0,00752 |
| 3 | **0,00021** | -0,00570 | 0,00611 | **-0,00044** | -0,00532 | 0,00428 | **0,00689** | 0,00129 | 0,01251 | **0,00576** | 0,00094 | 0,01050 |
| 4 | **-0,00105** | -0,00675 | 0,00472 | **-0,00123** | -0,00608 | 0,00348 | **0,00423** | 0,00040 | 0,00825 | **0,00450** | 0,00041 | 0,00897 |
| 5 | **-0,00552** | -0,01134 | 0,00014 | **-0,00082** | -0,00562 | 0,00397 | **0,00457** | 0,00048 | 0,00898 | **0,00505** | 0,00064 | 0,00961 |
| 6 | **-0,00259** | -0,00850 | 0,00314 | **-0,00138** | -0,00614 | 0,00328 | **0,00435** | 0,00046 | 0,00850 | **0,00531** | 0,00046 | 0,01052 |
| 7 | **-0,00636** | -0,01228 | -0,00069 | **0,00291** | -0,00181 | 0,00775 | **0,00424** | 0,00042 | 0,00845 | **0,00570** | 0,00063 | 0,01095 |
| 8 | **0,01853** | 0,01275 | 0,02450 | **0,00081** | -0,00413 | 0,00558 | **0,01049** | 0,00128 | 0,01986 | **0,00584** | 0,00065 | 0,01103 |
| 9 | **0,01350** | 0,00767 | 0,01949 | **0,00198** | -0,00292 | 0,00683 | **0,00524** | 0,00077 | 0,00985 | **0,00383** | 0,00036 | 0,00763 |
| 10 | **-0,00060** | -0,00649 | 0,00520 | **0,00338** | -0,00135 | 0,00822 | **0,00473** | 0,00051 | 0,00896 | **0,00430** | 0,00045 | 0,00846 |
| 11 | **0,00025** | -0,00570 | 0,00602 | **0,01009** | 0,00530 | 0,01516 | **0,00483** | 0,00064 | 0,00923 | **0,00764** | 0,00152 | 0,01366 |
| 12 | **-0,00468** | -0,01062 | 0,00099 | **0,00308** | -0,00167 | 0,00798 | **0,00482** | 0,00057 | 0,00924 | **0,00370** | 0,00030 | 0,00736 |
| 13 | **0,00095** | -0,00494 | 0,00675 | **-0,00322** | -0,00809 | 0,00152 | **0,00591** | 0,00061 | 0,01151 | **0,00408** | 0,00037 | 0,00807 |
| 14 | **-0,00310** | -0,00902 | 0,00249 | **0,00100** | -0,00384 | 0,00578 | **0,00438** | 0,00049 | 0,00855 | **0,00802** | 0,00205 | 0,01378 |
| Overall SD | **0,00872** | 0,00562 | 0,01367 | **0,00602** | 0,00370 | 0,00968 | **0,00533** | 0,00018 | 0,01138 | **0,00519** | 0,00038 | 0,01068 |

**Appendix A: Supplemental tables**

**Table A2.** Newman and Keuls' groups of populations of samples bulk density according to the laboratory that constructed them (sorted by decreasing mean bulk density). Lab number 15 represents the samples provided by UGent.

| Lab number | Mean (g.cm$^{-3}$) | SD (g.cm$^{-3}$) | Pop. size | NK Group |
|:---:|:---:|:---:|:---:|:---:|
| 1 | 1.8035 | 0.0094 | 5 | a |
| 2 | 1.7781 | 0.0141 | 5 | b |
| 8 | 1.7639 | 0.0494 | 5 | b c |
| 3 | 1.7551 | 0.0049 | 5 | b c d |
| 11 | 1.7540 | 0.0090 | 5 | b c d |
| 7 | 1.7528 | 0.0046 | 5 | b c d |
| 10 | 1.7425 | 0.0062 | 5 | c d |
| 12 | 1.7314 | 0.0168 | 5 | d |
| 4 | 1.6948 | 0.0198 | 5 | e |
| 13 | 1.6657 | 0.0291 | 5 | f |
| 5 | 1.6579 | 0.0133 | 5 | f |
| 6 | 1.6574 | 0.0177 | 5 | f |
| 9 | 1.6489 | 0.0056 | 5 | f |
| 14 | 1.6462 | 0.0136 | 5 | f |
| 15 | 1.6359 | 0.0113 | 14 | f |

**Table A3.** ID of Samples showing increasing SWRCs as function of the analysing laboratory and the round.

| Lab | Round 1 | Round 2 | Round 3 |
|---|---|---|---|
| 1 | / | / | / |
| 2 | / | / | / |
| 3 | 1, 2, 5 | 1, 2, 39, 40, 41, 42 | 1, 5, 81, 82 |
| 4 | / | / | / |
| 5 | / | 15, 17, 18 | / |
| 6 | 43 | / | 15, 16 |
| 7 | 56, 57, 59, 60 | 48, 56 | / |
| 8 | / | / | / |
| 9 | 10, 11 | / | 11 |
| 10 | / | / | / |
| 11 | 62, 63, 64, 65 | 61, 62, 75, 76, 77, 78 | 9, 61, 62, 65, 66 |
| 12 | / | 65, 66 | 25, 29, 30 |
| 13 | / | / | 63 |
| 14 | 21 | 20, 69, 70, 71, 72 | 23, 28 |

*Author contributions.* The conceptualization of the study was developed during the SOPHIE meeting of 2019 at Gembloux with the contribution of Hanane Aroui Boukbida, Gerben Bakker, Andrzej Bieganowski, Yves Brostaux, Wim Cornelis, Aurore Degré, Christian Hartmann, Krzysztof Lamorski, Axel Lamparter, Ana María Mingot Soriano, Attila Nemes, Martine van der Ploeg and Maarten Volckaert. Reference samples were provided by Wim Cornelis and Maarten Volckaert. All authors were involved in the data collection. The methodology for the data analysis was developed by Yves Brostaux, Aurore Degré, Benjamin Guillaume and Alexandre Pomes-Bordedebat. The manuscript was drafted and edited by Benjamin Guillaume. All authors were involved in the review of the paper.

*Competing interests.* The authors declare that they have no conflict of interest.

*Acknowledgements.* The rings for reference samples were provided by Royal Eijkelkamp.

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
