# Peer review of "Reproducibility of the Wet Part of the Soil Water Retention Curve : A European Interlaboratory Comparison"

_EGUsphere, 2022_

## Author Response (AR1)

**Reviewer 1 :**

We want to thank the reviewer for her/his comments on the manuscript. These elements, we believe, will definitely improve the article. We have added our response (in red) below each reviewer's comment (in bold).

This manuscript describes a well-performed interlaboratory comparison of measured water content in samples at low tensions, 10, 50, 100, and 300 hPa with standardized times of saturation, equilibration times, and oven drying time. Samples were composed of a mixture of glass beads and Portland cement with a supposedly stable structure, allowing repeated measurements on the same sample. Water retention was determined in 14 laboratories across Europe and samples moved among these laboratories in a predefined way, allowing the detection of inter- and intra-laboratorial variability or reliability of the determinations.

The experimental effort and the collected data are very interesting and show how imperfect the determination of the soil water retention property is. The statistics are presumably well-performed.

In the end, however, the authors leave the reader with some questions.

- **The main one: Which method/lab is the best one, and if we have so many differences between and within laboratories, how should we deal with that? What is the "correct" (or: reference) value we all should try to reproduce? To discuss this, I suggest an additional item at the end of the Discussion section, see my comments below (where I refer to the Conclusion section).**

The question that is being raised here is one that we believe to be fundamental and to which all of us aspire to an answer. To answer this question, a sample with a pre-established and previously well-known retention curve is required. Unfortunately, this was not the case with the samples we used. This was implicitly mentioned in lines 196-197 ("It should also be mentioned that the true value of the SWRC was unknown. Laboratories were compared according to their relative position with respect to the others and not against a fixed target value"). Therefore, based on the results from this trial, this question ("What is the correct value?") can only be answered by making quite strong assumptions. We would prefer to leave these questions open for further trials with new types of reference samples.

Nevertheless, as suggested, we have included lines 321-330 of the conclusion in a new item at the end of the Discussion section:

- **"4.5 A way forward to further improve the quality of SWRCs measurements**

The results presented in sections 3.3 and 3.4 show that inter- and intra-laboratory variability exists in the measurement of the SWRC. As discussed in Section 4.1, we suspect that some part of this variability can be attributed to the different methods and procedures used by each laboratory. Therefore, this variability could potentially be reduced by improving procedures and methods. Ideally, these should be adapted in such a manner that they allow the closest possible estimation of the "actual" SWRC of any soil sample. The prerequisite for this to be feasible is to have a fixed point of comparison; i.e. a reference sample whose retention properties are well known and remain relatively stable over time. However, these two requirements were not fulfilled in this study. Therefore, the way in which procedures and

methods should be adapted remains an open question. Nevertheless, based on the results of this study, we can suggest some general future directions to further improve the quality of the laboratory SWRCs measurements.

We believe that the reproducibility of SWRCs measurements within a same laboratory would be improved if a reference sample was used by each laboratory as an internal quality control. This implies that the SWRC of the reference sample must remain relatively stable over time, but its true value should not necessarily be known a priori. The reference samples used in this study are already used by one laboratory as an internal quality control. According to their experience, the samples remain fairly stable for about ten measurements as long as they are not oven dried.

Also, an important element that would allow to move forward would be to have a point of comparison. Thus, the emphasis should be placed on the development of reference samples for interlaboratory comparisons. Internal trials are underway to build reference samples with clays or sintered glass beads. Also, reference samples based on parallel capillary bundles with adjustable diameters could allow a "theoretically" reference retention curve to be calculated from Jurin's law. The feasibility of developing such a reference sample is also being explored.

The inter- and intra-laboratory variability could be reduced by improving and standardizing procedures and harmonizing methods and data. Ideally, all laboratories should endorse a unique Standard Operational Procedure for the same method and methods should be harmonized between each other. Improving and standardizing procedures requires a full assessment of the effect of each step of the procedures on the final SWRC measurement on a reference sample that fulfills the above-mentioned conditions (a priori known and relatively stable). Harmonization of methods can be achieved with interlaboratory comparisons on the same types of reference samples.

Finally, since procedures and methods could have an impact on the final measurement of the SWRC, the transparency of the procedures and methods used in SWRCs datasets should be ensured."

In my opinion, some issues need to be addressed by the authors:

- **The material (glass beads) are of sandy texture. There is no silt or clay at all in the samples. Such a pure sand sample will have a very steep retention and hydraulic conductivity curve. The steep K curve may lead to an easily occurring loss of hydraulic contact between the sample and porous medium or within the sample when submitted to suction or pressure. I would say, this kind of sample is among the most difficult to analyse. Wouldn´t the results be less pronounced in real soil samples containing clay, silt, and sand? In other words: didn´t you submit yourself to a very hard to analyse sample? This should be commented on.**

We generally agree with this reflection. The purpose of the article here is first and foremost to show that inter-laboratory variability exists (we believe regardless of the type of sample measured) and this should not be neglected. Nevertheless, we agree that some of the results presented here are not directly transposable to other types of samples. The idea was to have a sample that ensures that differences in weight between two suctions are significant (and thus are useful for internal quality

control). The used mixture (glass beads + cement) was selected after having tested several mixtures (with different materials but also different ratios of used materials).

To avoid any over-interpretation, we have added this disclaimer statement at the end of the 3.4 section (L182-184):

- "Results presented here are only representative of the reference samples that were measured in this particular case. Thus, estimates of inter- and intra-laboratory variability values are not directly transferable to other samples of different nature with different retention characteristics."
-
- **When reading line 297, I get confused about the method. From line 81, I assumed samples would be resaturated in between each tension (although that is not fully clear). But from this sentence in line 297, I conclude that unsaturated samples (equilibrated at the previous tension) were submitted to the next tension without rewetting? This needs a disambiguation. If this has been the applied protocol (i.e., not resaturating between tensions, but only wetting the suction plate before reallocating the samples), some hysteretic phenomena may also has been induced.**

Samples were not resaturated in between each tension. The choice of wetting the suction plate in between each tension (before reallocating the samples) was left to the laboratories. This will be clarified.

L88-89 :

- "At the beginning of each round, each sample was initially saturated for 48h. The mass of each sample was then measured after equilibration at different matric potential (or suction) values :…"
- **pF is defined in soil physics using units of energy/weight, a head unit, i.e., in cm (and not in energy/volume=pressure unit like Pa or hPa). pF is the value of log10(-h/cm), where h < 0 is the pressure head. See e.g. Koorevaar et al. p. 81 (P. Koorevaar, G. Menelik and C. Dirksen. 1983. Developments in Soil Science, 13. Elsevier.), or Lal & Shukla p.314 (R. Lal, M.K. Shukla. 2019. Principles of Soil Physics), or other textbooks on the subject. It is therefore odd to refer to a correction of cm to hPa (Table 1) or to refer to pF as the log10 of tension in hPa. The opposite would be correct. In line 93 you should write log10(h/cm) = pF, and in Table 1 you should refer to a correction of hPa to cm. As tension tables and sandboxes apply hanging water columns or a height above a water level, it is most likely that the laboratories effectively "measured" a vertical distance in cm, which could be converted to pF without any correction.**

This is true for sandboxes, but not for pressure plates. The latter is normally used with a pressure gauge and thus indicates pressures in hPa or bars. As the retention curve is composed using both methods, somewhere along the line cm's must be converted to hPa or vice versa. Indeed 1cm is not 1 hPa.

**Similarly, in lines 238-239 it is important to verify that those transforming their cm readings directly to pF are correct and that transforming hPa to pF (without correction) implies an error (and not the opposite, as suggested). This error is exactly quantifiable on the pF-scale, where it is equal to a constant value, exactly log10(hPa/cm) = log10(0.981) = 0.00833. In other words, when converting hPa to pF without correction, the assumed pF values will be 0.00833 units lower than the (correct) pF using cm units. This error does not depend on the pF itself and will therefore be relatively larger near saturation (when pF is low). It is not equal to 2% (as could be interpreted**

**from line 238). Furthermore, it will only affect the intercept, not the slope. I suggest you mention this all in your text.**

Yes, assimilating log10(hPa) as equal to pF is our mistake. This has been rectified in line with your recommendations. The notion of pF has been avoided in this article in favor of log10(ψ) and ψ is in hPa unit. This aims to be in line with the protocol which requested measurements to be made in the hPa unit (10 hPa, 50 hPa, 100 hPa and 300 hPa).

All "pF" units have been replaced by "log10(hPa)" in the main text, tables and figures. In equations, "pF" has been replaced by "log10(ψ)".

Results are presented and statistically analysed and compared, but in the Discussion section no convincing explanation is given for many of the observations. Maybe this is in the nature of the subject, and no conclusive explanations can be found for some of the findings, but there are many speculative affirmations, and some of them could be more elaborated, e.g.:

- **\*\* l202-206 – if possible, give the explanation Schelle te al. found for this difference. Why only on sandy soils?**

l202-206 have been replaced by:

L227-234 :

- "Nonetheless, Schelle et al. (2013) found that SWRCs measured with SP (suction plate) were less reproducible (wider spread) than those measured with the evaporation method in the pF range 0-2.5. This wider spread could be associated with contact issues with the plate and/or with the smaller sample size used with the SP method which could be smaller than the representative elementary volume of the soil. They also found that, for sandy soils, water contents are systematically smaller for SWRCs obtained with SP than with the evaporation method. These differences could be due to smaller initial saturation degrees of the sandy samples measured with the SP compared to those measured with the evaporation method. Temperature effects and dynamic non-equilibrium effects as found by Diamantopoulos & Durner (2012) could also have played a role in these differences."

- **\*\* l209-210 This is a very interesting observation. Elaborate on it more. In a pressure device, the pressure is applied via the air, while the sample bottom is at atmospheric pressure. In a suction device, the suction is applied by a hanging water column via a continuum of water-filled pores. So, the propagation of the applied tension is in the opposite domain (via air in one case, and liquid water in the other). That might impose differences, especially under very dry or very wet conditions.**

Thank you for the statement you have made here. It is much better explained than what we wrote in the article. If you agree, we would like to rephrase your words by mentioning that the atmospheric pressure is applied at the bottom of the plate instead of the bottom of the sample (it seems also important to consider that some labs use contact materials between the samples and the plate):

We have added :

L242 – 245 :

- "In a suction device, the suction is applied by a hanging water column via a continuum of water. In a pressure device, the pressure is applied via the air to the soil and plate water, while the

plate bottom is at atmospheric air pressure. So, the propagation of the applied tension/pressure might be different."

- **l211-214 Here it would be interesting to mention that, according to the Clausius-Clapeyron equation (assuming a temperature of 20 °C and relative humidity of 50% in the lab), an oven temperature of 100 °C corresponds to pF = 6.91; and 60 °C corresponds to pF = 6.65. So, your assumed "zero" water content corresponds to these two values of pF (might refer to https://doi.org/10.2136/sssaj1991.03615995005500040004x or https://doi.org/10.1002/saj2.20014 in this context)**

Yes, fully agreed.

L211-214 have been replaced by:

L246 – 257:

- "The procedures for the dry mass measurement of the samples may also have played a role in the observed differences. Indeed, the estimation of the intercept parameter of laboratory 8, which dried the samples at 100°C, was higher than the ones from the other laboratories, which dried the samples at 60°C (Fig. 6). This suggests that the dry masses measured by laboratory 8 were lower than those measured by the other laboratories. This may be explained by the fact that the assumed "zero" water content corresponds to two different water content values associated with two different water potentials. Indeed, the equilibrium potential of the water contained in a sample after drying depends on the drying temperature and the relative humidity of the air inside the oven (Ross et al. 1991). At liquid/vapor equilibrium and at constant relative humidity, the same sample dried at 60°C will have a higher water content than if it was dried at 100°C. This difference in water content will also depend directly on the shape of the dry part of the retention curve. Furthermore, for the same sample at the same initial condition, the drying time required to remove the same amount of water increases as the drying temperature decreases. If liquid/vapour equilibrium has not been reached after the prescribed time of drying, it is possible that the amount of water released is lower when drying at 60°C than at 100°C."

- **l218-219 It is hard to conclude this, as other factors varied between the laboratories as well.**

Indeed, we have to be more nuanced here. We replaced l218-219 by:

L261 - 263

- "From this study, it cannot be concluded that the use of contact materials did or did not improve the hydraulic contact between the samples and the porous plates. The results of laboratory 11 suggest that contact issues may have occurred even if filter paper was used as the contact material."

- **l236-238 The "pF" issue – see my previous comment.**

This has been rectified – see answer from the previous comment.

**l240-243 This difference can be perfectly quantified and corrected, so why not do that? For higher values of tension, this factor would become negligible, but at low suctions it could be very relevant. In my analysis, it would affect the intercept, not the slope.**

Indeed, this could easily have been corrected. Initially, we made this correction because we asked this information from all laboratories (reference level compared to the sample at which the suction is applied). However, in practice, this information is never transmitted from the laboratory to the end user. When pooling data from different laboratories using different reference levels to apply suction, this introduces variability, which in practice is never considered (even if it can be important at low suction). With this paragraph we wanted to shed light on this problem.

We have added:

L287-289 :

-   "This difference could have easily been corrected for after the measurement. However, in practice, this information is never transmitted from the laboratory to the end user. Hence, when pooling data from different laboratories using different reference levels to apply suction, this unknown difference introduces variability."

-   **\*\*l247 Lab temperature and humidity have a minor effect. I made a rough calculation and found that at RH=50%, the sensitivity of oven pF to temperature is of the order of -0.007/K. At 20 °C, the sensitivity to RH is -0.0002/%. In terms of water content, this is most probably negligible or even undetectable in a soil sample on a 0.01 g resolution balance.**

Yes, this statement has been removed.

-   **\*\*l275 Air humidity would not be likely to affect carbonation, as the relative humidity in the sample air will always be around 99-100%**

Yes, but only during the measurements. Samples are presumed to be dry when they are stored or shipped between two sets of measurements.

-   **\*\*l304-308 This seems very implausible to me. Why would this only occur at the highest tensions? And: how much weight gain could this represent? If you want to keep this paragraph, you should perform a calculation based on the mass of cement and the carbonation reaction involved in CO2 fixation.**

This paragraph has been deleted.

-   **The CONCLUSION section needs to be rewritten. The first part (l310-320) is a summary of the manuscript, where lines 315-320 contain some conclusions. Lines 321-330 are, in fact, a discussion about how to reduce the observed variability between measurements, and how to standardize. This would better fit in a new item at the end of the Discussion section, and could answer those questions I started with: which method is the best one, and how do we find the true value of water retention? This "best" method should not only allow better reproducibility among laboratories but (more importantly) it should be as close as possible to the true value.**

We have replaced the original conclusion with the following one:

-   **"5 Conclusion**

    Here, we presented an interlaboratory comparison of the measurement of the wet part of the SWRC conducted between 14 laboratories using artificially constructed, structured and porous samples as references. The experimental design combined with the data analysis procedure allowed the inter- and intra-laboratory variability to be revealed. Systematic differences in the

measurement of SWRCs attributed to laboratories resulted in a large interlaboratory variability. The variability explained by the differences between laboratories was more important than the variability explained by intrinsic differences between samples. The intralaboratory variability was laboratory dependent. The mean intralaboratory variability over all laboratories was approximately 45% smaller on the intercept and 15% smaller on the slope than the mean interlaboratory variability (Table A1). The samples slightly changed during the interlaboratory comparison, inducing variability which was part of our estimate of intra/interlaboratory variabilities. We believe that another part of the intra/interlaboratory variability can be attributed to the different methods and procedures followed by each laboratory that were not standardized. This calls for standardization of procedures and harmonization of methods.  This should be performed in the light of a fixed target value: a reference sample whose retention properties are well known and preferably remain stable over time. We believe that without such an effort, pedotransfer functions and large-scale maps of soil properties produced with databases constructed on multiple laboratories' inputs will keep carrying unknown levels of uncertainty and bias."

- **Some other (minor) comments:**
- **l18 – why not include transpiration? or "evapotranspiration"?**

The term "evaporation" has been replaced by "evapotranspiration".

- **l35 for completeness, here you should also add the field method using simultaneous measurements by tensiometers and water content sensors. See e.g. in https://doi.org/10.3390%2Fs21020447**

The following sentence on this in-situ method has been added.

L37-38

- "This method can also be implemented in situ using tensiometers and water content sensors installed side by side (Zeitoun et al., 2021)."

- **l56 add "measurement" - of measurement of soil hydrophysical properties**

This has been added.

- **l84 "ratio of fresh over dry masses" seems an incorrect description. Should be "ratio of water masses (the difference between wet and dry masses) over dry masses" or "ratio of wet over dry masses minus 1".**

Right! This has been corrected by :

L92-93 :

- "Gravimetric water content (wc in g.g−1) was calculated by the ratio of water masses over dry masses (water content = (fresh mass−dry mass)/dry mass)".

- **l90 A comment about why two missing curves?**

This is due to a mix-up by one laboratory in the samples exchange process. We did not want to encumber the article with this kind of not very useful statement.

- **l92: This should be the opposite: "A linear function was fitted to the measured wet part of SWRCs"**

Yes, we fully agree. l92 has been replaced by:

L102-103 :

- "To model our dataset, a linear function with log10(ψ) as the independent variable was adjusted to the measured wet part of SWRCs (Eq. 1)."

- **Equations 2, 3 and 4 – the "varying slope" -bracket should not include the (pF-1) term, but only the multiplying (slope coefficient) part.**

This has been changed according to what is prescribed here.

- **l124 (Figure 2) In the slope (beta-1) frequency distribution graph, you use the unit (g g-1 pF-1). pF is, in fact, a dimensionless quantity, and you might omit it. On the other hand, if you choose to include it, then your unit of density (y-axis) should be "pF"? Similar in Fig4, 5, ...**

According to the comment on pF units, slope units (g g-1 pF-1) have been replaced by g g-1 log10(hPa)-1.

- **l132 (Figure 3) It doesn´t seem correct to call these "SWRCs". The figure shows the four observed retention values connected by straight lines. This is better described by "observed water retention values at four pF values" or similar. And: Why not show the fitted lines instead of connecting lines?**

We agree with that. However, we would like to keep this figure with connecting lines to illustrate the section 3.6 (Increasing SWCRs). If the fitted lines are shown, some increases in water content between two increasing suction steps will be masked.

Nevertheless, we have added the colors (depending on the laboratory) to the individual SWRC parameter's distributions shown in Figure 4. This is equivalent to displaying the fitted lines (without having to add even more figures).

The caption of the figure 3 has been replaced by:

- "Figure 3: Observed series of four successive retention points (10 hPa, 50 hPa, 100hpa and 300 hPa) connected by straight lines. Water content is gravimetric (g g-1). One color represents one laboratory. This color code is kept constant throughout the paper."

- **Fig5: improve this caption. The "standard deviation of the two "STAY" samples" seems incomplete. Improve to e.g. "the standard deviation of the intercept and slope parameters of the two "STAY" samples"**

The caption of the figure 5 has been replaced by:

- "Figure 5: Densities of the posterior probability distribution of the varying intercept, z0n, and the varying slope, z1n, standard deviation of the two "STAY" samples of each laboratory (top row) and all laboratories together (bottom row)."

- **l150 0.1573 (comma to dot)**

This has been corrected

- **l150 replace "by the lab 1 and lab 14" by "by the lab 1 (highest bulk density) and lab 14 (lowest bulk density"**

This has been replaced.

- **l203 These acronyms are defined only in the Table 1 caption. I think it would be interesting to define them in the main text as well. The best place for that would be around line 31.**

These acronyms have been added in line 31.

- **l226 whole curve => (replace by) entire domain of pressure heads**

This has been replaced.

- **l228 is useful => are useful**

This has been corrected.

- **l237 "However, 1 cm of water column is not equal to 1 hPa but 0.98 hPa" => (replace by) "Units hPa and cm are commonly considered equivalent, but in fact 1 cm of vertical water column corresponds to 0.98 hPa"**

This has been replaced.

- **l278 seems => seem**

This has been corrected.

- **l281 I would remove "without a doubt" – that is a very strong affirmation.**

The "without a doubt" statement has been removed.

- **l304 delete "possible" and "probably" from this line.**

L303 - 308 have been deleted

- **l311 remove "for the first time"**

The "for the first time" statement has be removed in the new conclusion.

- **Table A2 (caption) add: "in order of decrease**

The caption of Table A2 has been replaced by:

- "Table A2. Newman and Keuls' groups of populations of samples bulk density according to the laboratory that constructed them (sorted by decreasing mean bulk density). Lab number 15 represents the samples provided by UGent."

**Reviewer 2 :**

We thank the reviewer for her/his comments on the manuscript. We have added our response (in red) below each reviewer's comment (in bold).

Manuscript 2022-1496 presents the outcomes of a European soil physics laboratory ring test investigating the reproducibility of water retention measurements for a tension range between 10 and 300 hPa. The authors find considerable differences in results that can be pinned down to different sample handling practices within the individual laboratories. The results imply that a more explicit definition of best practice rules is required for water retention curve measurements to minimize bias in the measurement results.

The here presented ring test is not the very first one undertaken to investigate measurement bias in water retention data, contrary to the authors' claim (L57). There was a very similar Swiss ring test undertaken several years ago with very similar results. However, a) the Swiss ring test is not published in a peer-reviewed journal and b) the here presented ring test has a wider scope in that it involves a large number of leading European soil physics research labs. The study is therefore very well suited for a publication in SOIL.

**I have several remarks that need to be carefully addressed by the authors before a publication. Most importantly, the Swiss ring test should be honored and shortly discussed, even if it is "only" published in the form of a report (Buchter et al., 2015; "Interlaboratory comparison of soil physical parameters"). This will not diminish the achievements of the present study but rather support the outcomes.**

We agree.

L57-60 have been replaced by:

L 59-67 :

- "To our understanding, no study other than that of Buchter et al. (2005) has carried out an interlaboratory comparison of SWRC measurements. This is partly due to the fact that an undisturbed soil sample cannot be transported from one laboratory to another and be measured several times without affecting the SWRC. Buchter et al (2015) circumvented this problem by using many samples from the same location and only using the samples in one round of SWRC measurement. They demonstrated that soil heterogeneity in the sampling area was negligible compared to the variability introduced by the different sample extraction, preparation and analysis procedures. However, this approach becomes very difficult to achieve when soil samples have to be transported by air and to countries where importing soil is restricted. Thus, in addition to innovation in measurement techniques, SOPHIE is working on the development of artificially constructed reference samples and the organization of interlaboratory comparisons, starting with the SWRC."

L192 have been replaced by:

L214-217 :

- "From saturation to drying, all laboratories used slightly different procedures. Similarly to the argument of Buchter et al. (2015) on the measurement of macropore volume, total pore volume and saturated hydraulic conductivity, we believe that procedural differences between laboratories can be at the origin of this interlaboratory variability."

After L206, we have added:

L234-240 :

- "Buchter et al. (2015) compared macropore volume (as equal to the difference between the total pore volume or water content at saturation and water content at a matric potential of 60 hPa) obtained with SB and with pressure cells on real soil samples. It can be observed that the volume of macropores determined by laboratories using the SB is generally greater with a greater variability than when determined by laboratories using the pressure cell. However, it is not clear from their results whether this difference is due to the method of obtaining the water content at 60 hPa (SB or pressure cell) or to the method of estimating the total pore volume (from bulk density, weighting at saturation, etc) which was also different between laboratories."

- We have removed in line 311: "for the first time".

I recommend minor revisions.

Specific comments:

- **L6-9: instead of bringing forward specifics of your study that are hard to understand without a proper introduction, better explain the type of statistics and the model you used to investigate the data.**

lines 6-9 have been replaced by:

L5-9 :

- "An inter-laboratory comparison was carried out between 14 laboratories, using artificially constructed, porous reference samples that were transferred between laboratories in according to a statistical design. The retention measurements were modelled by a series of linear mixed models using a Bayesian approach. This allowed the detection of sample-to-sample variability, interlaboratory variability, intralaboratory variability and the effects of samples changes between measurements."

- **L42-43: size effects are not "errors". Please rephrase.**

Yes, we agree. "errors" has been replaced by "variability".

- **L43: "pressure plates.." please explain better what you mean. You may also mention sand beds as they are connected with the same problematic. I also recommend including the sample saturation process at this point, as it is another part of the sample handling which can lead to different results between different labs.**

We agree, we have rephrased L43 by:

L49-51

- "Apparent hydrostatic equilibria (broken hydraulic contact and water flow being stopped before reaching hydrostatic equilibrium) might occur with Sand box, Sand/Kaolinite box, Suction plate or pressure plate methods, leading to overestimations of the water content, especially in the dry part of the SWRC."

We also believe that different sample saturation procedures could be at the origin of different results between different labs. However, we are not aware of any publication dealing with this issue to which we can refer.

- **L46: remove "In order to prevent hydrostatic non-equilibrium"**

This statement has been removed.

**Editor's comments :**

Dear Pr. Vanderborght.

We want to thank you for your comments on the manuscript. We have added our response below each of your comments (in bold).

- **I did not really understand how you defined the 'intralab' variability and whether the way you defined the intralab variability is correct. But, this is my interpretation and might be due to an incomplete or incorrect understanding. My interpretation of intralab variability would be the variability of measurements that are repeated in the same lab on the same sample. As far as I understand Figure 5, you used the standard deviation of the slope and intercept parameters of water retention curves that were measured three times on two samples that stayed in the same lab. The problem with this approach is that this standard deviation also includes the variation between the two samples that stayed in the same lab and the variation or uncertainty of the slope and intercept parameters due to a non-perfect linear relation between water content and pF. The latter uncertainty might be a consequence of uncertainty of measurements in a certain lab at a certain pF value but it might also be a consequence of the water-content-pF relation of the sample that is not linear. Therefore, I am not sure whether you can interpret this standard deviation as a variation that reflects the variation of a measurement that is repeated on the same sample in the same lab.**

The definition of intralab variability that is mentioned here refers to "measurement repeatability" as defined by the ISO/IEC Guide 99:2007:

"measurement precision under a set of repeatability conditions of measurement".

And "repeatability condition of measurement" can be defined as:

"condition of measurement, out of a set of conditions that includes the same measurement procedure, same operators, same measuring system, same operating conditions and same location, and replicate measurements on the same or similar objects over a short period of time" (ISO/IEC Guide 99:2007).

To our knowledge, there is no widely accepted definition of intralab variability. Therefore, we suggest to define intralab variability as the variability resulting from an intralaboratory comparison. "Intralaboratory comparison" is defined in ISO 17025:2017 as:

"organization, performance and evaluation of measurements or tests on the same or similar items within the same laboratory in accordance with predetermined conditions".

To our understanding, the "similar items" statement induces that samples must have similar properties. As we could not detect significant changes between rounds on "STAY" samples (we discuss this in l280), we can consider a "STAY" sample as a "similar item" between rounds.

Furthermore, the term "predetermined conditions", in our understanding, encompasses the same procedure that was given to all laboratories, but also the analysis of the data that was carried out in the same way for all laboratories. Therefore, for this study we suggest to define intralaboratory variability as:

"The variability between retention curves, modeled by linear regressions, measured on a similar ("STAY") sample within a same laboratory that uses a given measurement procedure."

We agree that there might be some variation between the two samples that stayed the same laboratory, and this may influence the estimate of intralaboratory variability. However, we believe that the way in which we determine the intralab variability minimizes the effect of possible variations between the two samples that stayed in the same lab. Indeed, the intralab variability associated with each of the two "STAY" sample is determined separately and are then merged together. In a way, this results in the variability that exists between the two samples not being captured.

In response to this comment, we have added in L151-157 :

- "From these results, one can determine the standard deviation of $z0n$ and $z1n$ for each of the "STAY" samples (between the 3 rounds). As each laboratory measured two "STAY" samples, an estimate of the intralaboratory variability of each laboratory can be made by pooling the density estimates of the standard deviation of the two samples (Fig. 5). As the intralaboratory variability associated with each of the two "STAY" sample is determined separately and are then merged together, this allows to minimize the effect of possible variations between the two "STAY" samples. Intralaboratory variability is therefore defined as the variability between retention curves, modeled by linear regressions, measured on a similar ("STAY") sample within a same laboratory that uses a given measurement procedure."

- **Another problem I have is with figure 10. It is not clear to me what figure 10 is actually showing. The caption says: joint distribution of intra-laboratory standard deviations on the intercept and slope. First, shouldn't it be joint distributions of intercept and slope?**

What is presented in figure 10 is basically the same as what is presented in the top row of figure 5. The only difference is that for the figure 10, posterior probability distribution of the varying intercept, $z0n$, and the varying slope, $z1n$, standard deviation of the two "STAY" samples of each laboratory are respectively represented on an x-axis and a y-axis (which refers to a "joint" distribution).

We have refined the caption of the figure 10 with the following one:

- "Joint posterior probability distribution of the varying intercept, $z0n$, and the varying slope, $z1n$, standard deviation of the two "STAY" samples of each laboratory (intralaboratory standard deviations)."

- **Second, are it the joint distributions of intercepts and slopes of all samples that were measured in a certain lab or only of the two samples that stayed in the same lab?**

Cf. answer above.

- **Third what do the ellipses actually represent? Are those contour plots of the bivariate normal probability density?**

Yes, these are 95% concentration ellipses assuming bivariate normality (this hypothesis is not true for lab 8, we agree). These are generated using the "ellipse_stat" function implemented in the "tidyverse/ggplot2" R package. The only purpose of these ellipses was to make the figure much more readable.

We have added at the end of the captions of figures 9 and 10:

- "The ellipses are for illustrative purposes only. Please, refer to table Table A1 for estimates of interlaboratory and intralaboratory variabilities.".

- **Fourth, you should check the colors used in the figure and in the text referring to lab 11 and 14. In the text you refer to red and blue ellipses whereas in the figure it are pink and purple ellipses.**

Yes, we have corrected that.

I think you can address these comments by making some clarifications in the text on what you actually mean with 'intralab variability' and what it represents.